# Glutathione Supplementation of Parenteral Nutrition Prevents Oxidative Stress and Sustains Protein Synthesis in Guinea Pig Model

**DOI:** 10.3390/nu11092063

**Published:** 2019-09-03

**Authors:** Guillaume Morin, Clémence Guiraut, Marisol Perez Marcogliese, Ibrahim Mohamed, Jean-Claude Lavoie

**Affiliations:** 1Department of Nutrition, Université de Montréal, 2405 Chemin de la Côte-Sainte-Catherine, Montréal, QC H3T 1A8, Canada; 2CHU Sainte-Justine Research Centre, Université de Montréal, 3175 Chemin de la Côte-Sainte-Catherine, Montréal, QC H3T 1C5, Canada; 3Department of Pediatrics-Neonatology, CHU Sainte-Justine, Université de Montréal, 3175 Chemin de la Côte-Sainte-Catherine, Montréal, QC H3T 1C5, Canada

**Keywords:** parenteral nutrition, glutathione supplement, parenteral cysteine, pro-cysteine, protein synthesis, glutathione, redox potential

## Abstract

Peroxides contaminating parenteral nutrition (PN) limit the use of methionine as a precursor of cysteine. Thus, PN causes a cysteine deficiency, characterized by low levels of glutathione, the main molecule used in peroxide detoxification, and limited growth in individuals receiving long-term PN compared to the average population. We hypothesize that glutathione supplementation in PN can be used as a pro-cysteine that improves glutathione levels and protein synthesis and reduces oxidative stress caused by PN. One-month-old guinea pigs (7–8 per group) were used to compare glutathione-enriched to a non-enriched PN, animals on enteral nutrition were used as a reference. PN: Dextrose, amino acids (Primene), lipid emulsion (Intralipid), multivitamins, electrolytes; five-day infusion. Glutathione (GSH, GSSG, redox potential) and the incorporation of radioactive leucine into the protein fraction (protein synthesis index) were measured in the blood, lungs, liver, and gastrocnemius muscle. Data were analysed by ANOVA; *p* < 0.05 was considered significant. The addition of glutathione to PN prevented the PN-induced oxidative stress in the lungs and muscles and supported protein synthesis in liver and muscles. The results potentially support the recommendation to add glutathione to the PN and demonstrate that glutathione could act as a biologically available cysteine precursor.

## 1. Introduction

Parenteral nutrition (PN) is essential for many patients with different gastrointestinal diseases. The causes are variable, ranging from intestinal insufficiency to the need for nutrition support in clinical conditions for which enteral nutrition is not indicated or limited [1]. This mode of nutrition results from advanced technologies that allow all nutrients to be in the same solution and administered intravenously. However, these nutrients are reactive and some of them give electrons to dissolved oxygen, generating oxidative molecules [2,3,4,5,6,7]. Among these oxidative molecules are the peroxides that were shown to induce oxidative stress and the loss of pulmonary alveoli in a newborn animal model [8]. They were also associated with bronchopulmonary dysplasia, characterized by low pulmonary alveolarization in premature infants receiving PN [9]. These consequences are explained by the low capacity of newborns to detoxify peroxides by glutathione peroxidases. While their levels of these enzymes are adequate [9], the level of glutathione is too low [9,10]. This defect is caused by an insufficient availability of cysteine to support glutathione synthesis [2,11]. In addition to its cofactor function for glutathione peroxidases, glutathione has also been known for a long time to be a pool of cysteine for the body [12,13]. This particular role of glutathione was demonstrated by enriching PN with glutathione [14], which prevented oxidative stress and alveoli loss in the lungs of newborn animals on PN [15]. These experiments suggest that PN, even if it contains cysteine, leads to cysteine deficiency.

The mechanism of action of glutathione enrichment is explained as follows. During the detoxification of peroxides by glutathione peroxidases, the reduced form of glutathione (GSH) is transformed into its disulfide form (GSSG). The concentration ratio of GSH on GSSG strongly influences the redox status of the cells, which is reflected by the redox potential of glutathione calculated by the Nernst equation. High levels of GSSG or low levels of GSH lead to oxidative stress (increased redox potential). Glutathione peroxidase activity is directly influenced by the availability of GSH, whereas GSH synthesis is limited by the availability of cysteine [16]. A main physiological way to improve the level of cysteine in the tissues is to use plasmatic glutathione [17,18,19]. γ-Glutamyltransferase transfers the γ-glutamyl moiety of glutathione to another circulating amino acid. The two formed dipeptides, the γ-glutamyl-amino acid and the remaining cysteinylglycine of glutathione, are taken up by the cells where the amino acids are cleaved by dipeptidases [17]. Glutathione in the plasma comes from the liver, from which it is actively exported [20]. The high capability of the liver to synthetize glutathione comes from its efficiency in transforming methionine into cysteine by the transsulfuration pathway. The immaturity of cystathionase (last enzyme of this cascade) in premature infants [21] could explain the low glutathione levels in this population. In addition, the peroxides generated in the PN solution inhibit methionine adenosyl transferase [11] (the first enzyme of this same cascade). Thus, PN infusion results in decreased levels of GSH in the lungs and blood of animals, exacerbating oxidative stress [8,15]. The cystathionase activity increases with age [22] whereas the impact of peroxides seems independent of age. Indeed, in children (aged 1 to 10 years) receiving home-PN, the level of glutathione in blood is equal to half that of control children [23].

Studies have suggested that the cysteine content in PN was not sufficient to support glutathione synthesis. Since the affinity of cysteinyl-tRNA synthetase for cysteine (Km = 1–20 µM [24]) is about 30 times greater than that of γ-glutamylcysteine ligase (Km = 200–350 µM [25,26]), the first enzyme for the synthesis of GSH, cysteine in PN seems to be used preferably for protein synthesis, especially in growing individuals. However, unlike preterm newborns who receive short-term PN for a limited time, studies reported that infants receiving long-term parenteral nutrition had a lighter weight [27] and a smaller height [23,27] than the population of the same age. We hypothesize, that PN does not provide enough cysteine to support glutathione and protein synthesis.

As it was suggested to enrich PN with glutathione in order to prevent the development of chronic pulmonary complications in premature newborns [2], it remains relevant to explore, in a model of a growing animal receiving PN, the capacity of supplemented glutathione to prevent oxidative stress and improve protein synthesis in various organs. Therefore, growing animals (one-month-old guinea pigs) have been used to compare the impacts of enriched or unenriched PN with glutathione, in relation to orally fed animals used as a reference group. Glutathione (GSH, GSSG, and redox potential) values and the incorporation of radioactive leucine into the protein fraction (as an index of protein synthesis) were measured in the blood, lungs, liver, and gastrocnemius muscle. The study shows that the addition of glutathione in PN can be used to prevent PN-induced oxidative stress, as has occurred in the lungs and muscles, and to sustain protein synthesis in organs with a greater need for protein synthesis during growth, such as the liver and muscle. The results may lead to a subsequent clinical recommendation to supplement PN with glutathione as pro-cysteine.

## 2. Methods

### 2.1. Experimental Design

Twenty-four one-month-old male Hartley guinea pigs, weighing 288 ± 3 g (Charles River Laboratories, Saint-Constant, QC, Canada), were housed in the animal facility for 5 days for acclimation (23–25 °C; 12:12 h light:dark). The PN animals were anaesthetized (87 mg/kg ketamine + 13 mg/kg xylazine and isoflurane gas for maintenance) in order to insert a catheter in the external jugular vein [8,11,15,28]. During the first two days after surgery, the animals received through the catheter 0.9% (w/v) NaCl containing 1 IU/mL of heparin i.v. and the infusion rate was gradually increased from 0.5 to 1.5 mL/hr. During this time, the animals were fed ad libitum with regular guinea pig food and had free access to tap water. The average daily caloric intake is described in Table 1. Animals who recovered 90% of their initial weight were included in the 5-day exclusive PN protocol.

The representation of our study groups in a clinical situation includes: The PN with no GSSG supplementation that represents the “standard of care or control group”, the PN with GSSG supplementation that represents the “intervention group”, and the ad libitum animal group that represents healthy individuals “reference group”:(1)Reference group: Animals of the same age without any manipulation and fed with regular guinea pig food.(2)PN: Animals exclusively fed by PN, having free access to tap water. The PN was compounded with 10% (w/v) glucose, 2% (w/v) amino acids preparation (Primene, Baxter, Toronto, ON, Canada), 2% (w/v) lipid emulsion (Intralipid 20%, Fresenius Kabi, Mississauga, ON, Canada), electrolytes, 1% (v/v), multivitamin preparation (Multi-12, Sandoz, Boucherville, QC, Canada), and 1 U/mL heparin. PN solutions were freshly prepared daily and gradually administered at an average rate of 129 mL/kg/day, giving an average caloric intake of 85 kcal/kg/day. These values are close to the recommendations in paediatric parenteral nutrition [29].(3)PN + 10 µM GSSG: Intervention group, animals receiving PN enriched with 10 µM GSSG. GSSG was used because it has a better stability in the PN solution than GSH [30] and because its affinity for γ-glutamylcysteine transferase is similar to that of GSH [31]. The chosen concentration is the same as previously used with success to prevent pulmonary oxidative stress in neonatal guinea pigs [15].

After 5 days, the animals were sacrificed. Blood, liver, lungs, and gastrocnemius muscles were collected, processed, and kept at −80 °C until the assays. Plasma was obtained following blood centrifugation at 7200× *g* for 4 min.

In accordance with the principles of the Canadian Council on Animal Care (CCAC), the Institutional Committee for Good Practice with Animals in Research of the CHU Sainte-Justine approved the protocol.

### 2.2. Determinations

Peroxide concentrations in the PN solutions were assessed by the FOX assay based on the colorimetric reaction (560 nm) between xylenol-orange and ferric iron generated after oxidation of ferrous iron by peroxides [32]. The measurements were made after a 3-h incubation and H_2_O_2_ was used for the standard curve [23,28].

The histological determination of hepatic steatosis was done using red oil O staining. Frozen livers embedded in a cryomedium were cut 5 μm thick, placed on glass slides and kept at −80 °C until their staining. Slides were air-dried for 60 min, then placed in consecutive baths of 10% formalin in PBS (10 min), running distilled water (2 × 3 min), propylene glycol 100% (10 min), red oil O solution (60 min, ab150678, abcam ON, Canada), propylene glycol 85% (2 × 3 min), running distilled water (2 × 3 min), Mayer’s hematoxylin (1 min 30), running tap water (2 min), and finally dipped in distilled water (3 × 8 dips). Slides were kept in distilled water until they were mounted with an aqueous medium.

The red oil O staining was blindly assessed. Two images at two different locations were acquired for each liver under 20× magnification. Using ImageJ, each image was converted from RGB to CMYK. Cyan and magenta channels were extracted, transformed into grayscale image and then subtracted from each other (Cyan subtracted to Magenta) to subtract the background from the lipid staining. The resulting images were thresholded (24–255), and the stained area as well as the particles size (Size: 0–200 μm, Circularity: 0.50–1.00) were measured.

For the determination of GSH and GSSG, immediately after sampling, the tissues were homogenized in 5% (w/v) freshly prepared metaphosphoric acid (in 5 volumes for liver, lungs, and muscle samples, and in 3 volumes for the whole blood), and centrifuged for 4 min at 7200× *g*. GSH and GSSG in supernatants were measured by capillary electrophoresis/UV according to the previously published method [8,11,15,28] while protein levels were measured in pellets by the Bradford method. The redox potential was calculated using the Nernst equation.

The low concentration of glutathione in the plasma (µM) does not allow its determination by capillary electrophoresis. An enzymatic method based on the reduction of DTNB by GSH, generating a compound absorbing at 412 nm, was used [33,34]. The system includes the regeneration of GSSG into GSH by glutathione reductase + NADPH. Therefore, the increased absorbance over time is proportional to the level of GSH + GSSG in the sample. The standard curve was performed with GSSG and the results were reported as total glutathione expressed in GSH equivalent (1 GSSG = 2 GSH).

The protein synthesis index was evaluated by measuring the incorporation of ^3^H-L-leucine [35,36] into the tissue on days 3 and 4, at times where the daily caloric intake was constant (Table 1). At each of these days, 100 µCi of ^3^H-L-leucine was added to the PN. The last day of infusion was without radioisotope. Five hundred mg of tissue were homogenized with 5 volumes of 5% (w/v) metaphosphoric acid and centrifuged at 13,000× *g*/20 min/4 °C. Radioactivity (dpm) was measured by scintillation in the supernatant and the pellet (protein fraction).

The determination of haemoglobin and plasma urea values were utilized to assess the animal’s overall health, including the presence of anemia, dehydration, and starvation. The plasma concentration of urea was measured by the method of Fearon [37] reviewed by Rahmatullah and Boyde [38]. The method was based on a colorimetric reaction (520 nm) following the interaction between urea and diacetyl monoxime in the presence of thiosemicarbazide. The results were extrapolated from the standard curve generated with urea. The haemoglobin level assessment was based on the oxidation of haemoglobin to methaemoglobin in the presence of ferricyanide. The complex absorbed at 540 nm. A commercial kit (B4184, Sigma-Aldrich, Saint-Louis, MO, USA) was used.

### 2.3. Statistical Analyses

All data are presented as mean ± S.E.M. The groups were compared by ANOVA, using orthogonal comparisons, after verifying the homoscedasticity by Bartlett Chi^2^. Mean daily caloric intake data were logarithmically transformed to satisfy the homoscedasticity. Pearson’s correlations were used to quantify the weight gain of animals over of the 5-day duration of the experiment. The significant threshold was set at 0.05.

## 3. Results

### 3.1. Animal Characterization

Over the course of PN, one animal was removed from the experiment due to occlusion of the jugular catheter. Thus, 23 animals were included in this study.

The daily caloric intake (Table 1) increased over time to reach a plateau on the third day. There was no statistical difference between day 1 and 2 (F_(1,65)_ = 1.7) or between days 3, 4, and 5 (F_(1,65)_ < 2.3). A statistical difference was documented between days 1 and 2 and between days 3,4, and day 5 (F_(1,65)_ = 28.2, *p* < 0.0001).

Bodyweight (Figure 1) of the reference group increased by 14% during the five-day experiment (including all animals in this group, y = 9.7 g·day^−1^ + 352 g; *r*^2^ = 0.64, *p* < 0.01), while it decreased over time in the PN group (including all animals in this group, y = −3.6 g·day^−1^ + 329 g; *r*^2^ = 0.18, *p* < 0.01), and in PN + 10 µM GSSG group (including all animals in this group, y = −3.8 g·day^−1^ + 334 g; *r*^2^ = 0.18, *p* < 0.02). The slopes and the 6% decrease over time were similar between the PN groups. Bodyweight at day 0 differed between the reference group and the PN groups (F_(1,20)_ = 15.3, *p* < 0.01), they were similar between the PN group and the PN + 10 µM GSSG group (F_(1,20)_ = 0.8).

PN had induced hepatic steatosis (Figure 2). This is a well-known complication of PN [39]. The number of lipid droplets was the lowest in the reference group (F_(1,18)_ = 55.25, *p* < 0.0001) compared to PN ± GSSG groups (Figure 2). It was higher in the PN + GSSG group than in the PN group (F_(1,18)_ = 8.13, *p* < 0.05). The lipid droplet average size was higher in the PN ± GSSG groups (F_(1,18)_ = 12.27, *p* < 0.01). The impact of GSSG supplementation on the size of lipid droplets did not reach the statistical significance (F_(1,18)_ = 3.23) (Figure 2).

Comparisons of haemoglobin and plasma urea values (Table 2) assessed the general health status of the animal, such as anemia, dehydration, and starvation. At day 1, the haemoglobin concentrations did not differ between groups (F_(1,36)_ < 2.9). At day 5, they were similar between reference group and PN + GSSG group (F_(1,36)_ = 2.00) and were lower than those in the PN group (F_(1,36)_ = 13.24, *p* < 0.01). Plasma urea concentrations on the last day of the experiment were not significantly different between the groups (F_(1,20)_ < 0.1).

### 3.2. Oxidative Stress

The PN solutions of the present study were contaminated with 272 ± 14 µM peroxide. This concentration was similar to levels previously measured in such solutions [3,28]. This amount of peroxide has the potential to induce oxidative stress.

Blood: Addition of glutathione in PN did not induce, in the whole blood, changes in GSH (7.3 ± 0.4 nmol/mg prot), GSSG (0.22 ± 0.02 nmol/mg prot), redox potential (−198 ± 1 mV) or proportion of the oxidized glutathione (5.6 ± 0.3%). All were similar to reference animals (F_(1,19)_ < 3.8). In plasma, the total glutathione concentration was similar (F_(1,19)_ = 0.7) between the PN and PN + GSSG groups (1.8 ± 0.1 μM), lower (F_(1,19)_ = 40.6, *p* < 0.0001) to the value measured in the reference group (2.9 ± 0.1 μM).

Lungs: The low level of GSH (F_(1,20)_ = 6.53, *p* < 0.02) was corrected by the addition of GSSG in PN, which was similar (F_(1,20)_ = 0.7) to that measured in the reference group (Figure 3A). The GSSG values (Figure 3B) did not differ between the groups (F_(1,20)_ < 2.2). With a greater proportion of glutathione in the oxidized form (Figure 3D), relative to the reference group (F_(1,20)_ = 4.07, *p* = 0.057), the PN group had the most oxidized redox potential (F_(1,20)_ = 5.88, *p* < 0.05). Redox in PN + GSSG was similar to the reference group (F_(1,20)_ < 0.3) (Figure 3C).

Liver: There was no statistically significant difference (F_(1,20)_ < 2.4) between groups for GSH, GSSG, redox potential or proportion of glutathione in the oxidized form (Figure 3).

Muscle: GSH levels (Figure 3A) did not differ between groups (F_(1,20)_ < 3.5). The high level of GSSG (Figure 3B) in the PN group was corrected by the addition of GSSG in PN (F_(1,20)_ = 25.72, *p* < 0.0001), the PN + GSSG group value was similar to that measured in the reference group (F_(1,20)_ = 0.19). With the largest (F_(1,20)_ = 14.9, *p* < 0.001) proportion of oxidized glutathione (Figure 3D), the PN group had the most oxidized redox potential (F_(1,20)_ = 8.40, *p* < 0.01). Redox (F_(1,20)_ = 3.43), as well as the proportion of glutathione in the oxidized form (F_(1,20)_ = 0.68) in the PN + GSSG group was similar to the value in the reference group (Figure 3C).

### 3.3. Protein Synthesis Index

Blood: Although the total protein contents of whole blood in the three groups were not different (149 ± 3 mg/mL; F_(1,20)_ < 1.8), they were higher (F_(1,20)_ = 15.4, *p* < 0.001) in the plasma of animals receiving PN + GSSG and in the reference group (60 ± 1 mg/mL; F(_(1,20)_ = 0.2) than in the plasma of the PN group (52 ± 2 mg/mL).

Lungs: There was no difference in protein content (Figure 4A) between groups (F_(1,20)_ < 1.8). The protein synthesis index (Figure 4B) in the PN group was not modified by enrichment of PN with GSSG (F_(1,13)_ < 0.8).

Liver: The protein content (Figure 4A) was higher (F_(1,20)_ = 9.30, *p* < 0.01) in the reference group compared to the PN and PN + GSSG groups. Although the addition of GSSG to PN did not influence the protein level (F_(1,20)_ < 0.1), it improved the protein synthesis index (F_(1,13)_ = 9.05, *p* < 0.01) (Figure 4B).

Muscle: Although there was no difference in protein content (Figure 4A) between groups (F_(1,20)_ < 0.1), the protein synthesis index (Figure 4B) was improved (F_(1,13)_ = 9.60, *p* < 0.01) by the supplementation of PN with GSSG.

## 4. Discussion

The study shows that the addition of GSSG in PN may be used to prevent oxidative stress induced by PN, as has occurred in the lungs and muscles, and to support the protein synthesis in organs with the greatest need for the synthesis of protein during growth such as the liver and muscles. Even a further dose-response study is required to confirm these effects, the results already support the hypothesis that PN, as administered in a clinical setting, could be suboptimal in providing the amount of cysteine needed by the body and that GSSG supplementation could be used as pro-cysteine to support its biological activity.

The design of the experimental protocol consisted of evaluating the impacts of GSSG supplementation of PN on oxidative stress and protein synthesis. Thus, the PN group can be considered as the control group. In order to appreciate the magnitude of the effect of the GSSG supplementation, animals without manipulation served as a reference group and not a control group. Indeed, there was a significant dissimilarity between the PN groups and this reference group. A main difference concerned metabolizable nutritional intakes. After an acclimatization period for all animals, the caloric intake of animals receiving PN increased gradually to reach a plateau after three days. This progression constituted a specific period of acclimatization. In addition, surgery to fix the catheter is a metabolic stress as demonstrated by a lower body weight of 5% to 7% relative to the reference group after two days of ad libitum feeding with regular guinea pig food, before PN administration.

In this guinea pig model, five consecutive days of PN resulted in ~6% weight loss in all PN (±GSSG) animals, compared to 14% weight gain in the orally fed reference animals. These observations are consistent with previous studies using guinea pigs [40] or rats [14] as a PN model. One could argue that a nutritive deprivation would have led to catabolism. Urea is then measured in plasma since it is the main nitrogen product of protein degradation. Here, the similarity of urea plasma concentrations between groups does not support the presence of catabolism. In addition, equal levels of protein per gram of muscle tissue in all groups, including reference group, suggest that the PN animals did not suffer from inadequate caloric intake leading to higher protein catabolism. Despite an apparently adequate caloric intake, this lack of weight gain could be explained by qualitative and quantitative differences in the metabolizable nutrient intake between PN groups and the reference group, or by the impact of the nutrient delivery route during growth. Another possibility is that this lack of growth in PN animals could be explained, at least in part, by a partial inability to de novo synthesize proteins. Low glutathione levels in the blood of premature infants [9] and children with chronic PN [23], combined with data indicating a shorter height in children on chronic PN [23,27] support the hypothesis of a non-optimal availability of energy and/or cysteine leading to suboptimal protein synthesis. Energy is an important factor for this synthesis. PN groups received energy intakes similar to those recommended in the recent (2018) ESPGHAN/ESPEN/ESPR/CSPEN guidelines for preterm infants [29]. However, this recommendation does not take into account the energy required for thermoregulation, and the basal metabolism and activity (~50 kcal/kg/day) as suggested by Reichman BL et al. [41]. The energy cost of these parameters is unknown in our animal model. Suboptimal energy intake can compromise protein synthesis. On the other hand, it may not be appropriate to compare the energy requirements of a month-old guinea pig to a premature newborn. The second important factor is the availability of amino acids, including cysteine. Here, GSSG has been used as pro-cysteine.

The presence of cysteine in some parenteral amino acid preparations, as here with Primene, is controversial. In the presence of an oxidant, such as dissolved oxygen in a parenteral solution, cysteine oxidizes rapidly to cystine, whose solubility is about 1500 times lower (Drugbank, www.drugbank.ca). Primene contains 189 mg cysteine/ L (Product monograph, Primene 10%, Baxter Corporation, Mississauga, ON, Canada, 2015). The solubility of cystine is 190 mg/L (Drugbank). Thus, as a precautionary measure, it may be unwise to increase the concentration of cysteine in amino acid preparations. N-acetyl-cysteine has been proposed to enrich parenteral nutrition. However, a Cochrane meta-analysis found no significant effect of using N-acetyl-cysteine in order to improve low glutathione levels in premature infants and to reduce the incidence of several complications related to prematurity [42]. This report casts doubt on the usefulness of N-acetyl-cysteine in PN. It is possible, however, that this is specific to premature neonates. On the other hand, the usefulness of a glutathione supplement as a precursor of cysteine appears to be independent of age. Here, even with a concentration of 10 µM (less than one percent of the cysteine concentration initially dissolved in the PN), the GSSG added to the PN shows a biological availability of cysteine. However, the lack of correction of total glutathione in the plasma of the PN + GSSG group suggests that a concentration of 10 µM may not be optimal. Further studies need to be initiated to determine the optimal concentration of GSSG to be added to the PN before considering its clinical use.

The impact of GSSG supplementation was expected to vary according to the organ. Beyond comparisons between groups, the figures illustrate the high variability between studied organs for both glutathione values and protein levels, suggesting different cysteine requirements. The liver contains the highest level of GSH. It is the only organ that actively exports GSH to plasma. Thus, its glutathione synthesis rate is high and depends mainly on the transformation of methionine into cysteine. With a redox potential of <−240 mV, the hepatic cells are proliferating [43]. Thus, protein synthesis should be high. The protein synthesis index is improved in the presence of GSSG in PN. Since glutathione data were reported as a function of protein content (GSH, GSSG) or volume (redox potential), a confounding bias in comparison with the reference group could be the presence of lipids in the PN groups (Figure 3). This bias could also explain the lower protein content in the animals of the PN groups (Figure 4A). Steatosis is well known to be a PN-related complication [39]. Here, the lungs are the next organ in glutathione content. It actively exports GSH into the pulmonary lining fluid [44]. The high synthesis of glutathione in the lungs depends on the presence of glutathione in the plasma. The action of γ-glutamyl transferase allows the release of cysteine into the cells. With a normal redox potential of about −220 mV, the lung cells are proliferating [43]. However, the peroxides contaminating the PN [2,3] induce a lower level of GSH and, consequently, a shift of the redox potential towards a more oxidative state, associated with the cellular differentiation [43]. Therefore, PN can have an impact on lung development. The addition of GSSG in PN prevented this change. Of the three organs studied, the muscle is the one with the lowest GSH level, at one-sixth of the amount measured in the liver. With a redox potential of ~ −210 mV, the muscle cells are at the limit between proliferation and differentiation [43]. Thus, with a redox potential of about −200 mV in the PN group, the cells are more differentiated. These data suggest that here too, the PN can have an impact on muscle development. In contrast to the lungs, the oxidation of the redox potential in the PN group is caused by a higher value of GSSG, suggesting the presence of high levels of peroxides. Oxidized redox and high GSSG value are associated with abnormal calcium metabolism [45,46,47,48]. By preventing oxidative stress, a supplement of GSSG could preserve muscle function. Since the need for glutathione synthesis was relatively lower, the protein synthesis index was higher in the PN+GSSG group. In whole blood, PN and PN + GSSG did not influence glutathione concentrations or the redox potential. This highlights the difficulty for clinical studies in using the values measured in the whole blood as a reflection of glutathione in different organs. On the other hand, data on glutathione and protein concentrations in the plasma suggest that the glutathione levels in the plasma are the first to be depleted in glutathione deficiency and the last to be normalized (after fulfilling all needs of different organs). Hence, it is sensitive and specific measure for deficiency but less sensitive with glutathione repletion. One of the main limitations of the study is the use of a single concentration of GSSG. The choice of this concentration of 10 μM was based on a previous study showing the correction of plasma glutathione levels and the prevention of oxidation of the pulmonary redox potential in newborn guinea pigs [15]. One-month-old animals might need more GSSG. Others [14] reported that the infusion of a high dose of GSSG (8.9 mM) in rats (~6 weeks-old) receiving PN improved the plasma cystine concentration; the plasma concentration of glutathione was not reported. Here, using 1000 times less GSSG (10 μM), it is likely that we cannot measure a difference in plasma cysteine or cystine levels. At this relatively low concentration, circulating glutathione is used to enrich the cells, not the plasma, in cysteine. Moreover, the low level of plasma glutathione was not corrected by the addition of GSSG in PN, suggesting suboptimal supplementation. Nevertheless, the present study is a proof of concept of the bioavailability of GSSG as a precursor of cysteine in growing guinea pigs that are nourished by PN.

Steatosis was expected in the PN groups, but it was surprising to observe more lipid droplets in the PN group containing GSSG than in the PN group without GSSG. The size of the droplets was not influenced by the presence of GSSG in the NP. The direct impact of hepatic glutathione (GSH or GSSG) or redox potential was excluded because they were not influenced by PN or PN + GSSG (Figure 3). A future dose-response study will confirm whether the difference observed is due to a statistical error of Type 1 or whether supplementation with GSSG induces or aggravates NP-induced steatosis.

In view of a possible clinical application, a subsequent study should document the effect of increasing doses of GSSG on lipid accumulation in liver as well cysteine/cystine plasma concentrations in addition to glutathione levels and protein synthesis index. With a half-life of approximately 15 min, as demonstrated in humans [49], saturation of plasma glutathione concentration could be the indicator of the optimal dose of GSSG to be added to the PN.

## Figures and Tables

**Figure 1 nutrients-11-02063-f001:**
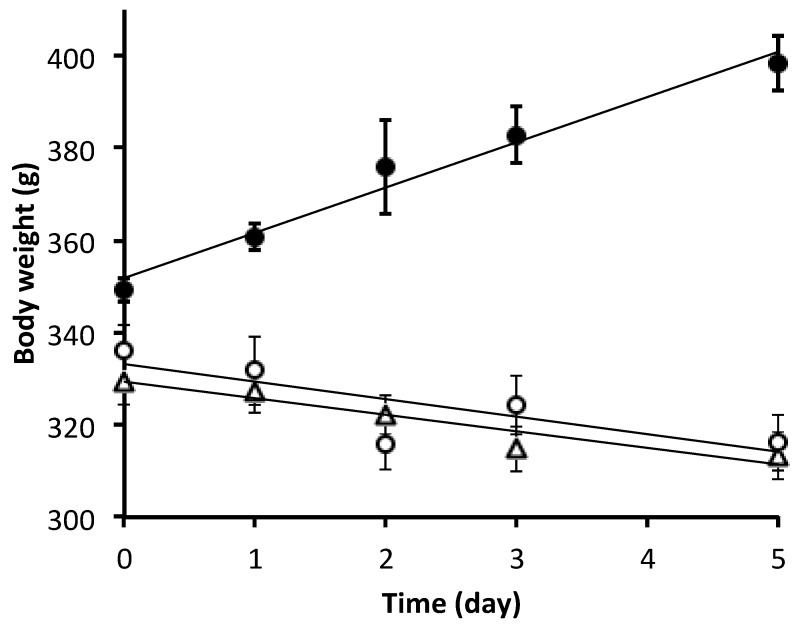
Body weight over the duration of the experiment. Data are presented as mean ± S.E.M. (*n* = 6–8/group/day). The equations of the linear curves were obtained by taking into account all the animals of the concerned group. Dark circle: reference group; y = 9.7 g day^−1^ + 352 g, *r*^2^ = 0.64, *p* < 0.01. Open triangle: parenteral nutrition (PN) group; y = −3.6 g day^−1^ + 329 g, *r*^2^ = 0.18, *p* < 0.01. Open circle: PN + 10 µM disulfide glutathione (GSSG) group; y = −3.8 g day^−1^ + 334 g, *r*^2^ = 0.18, *p* = 0.02.

**Figure 2 nutrients-11-02063-f002:**
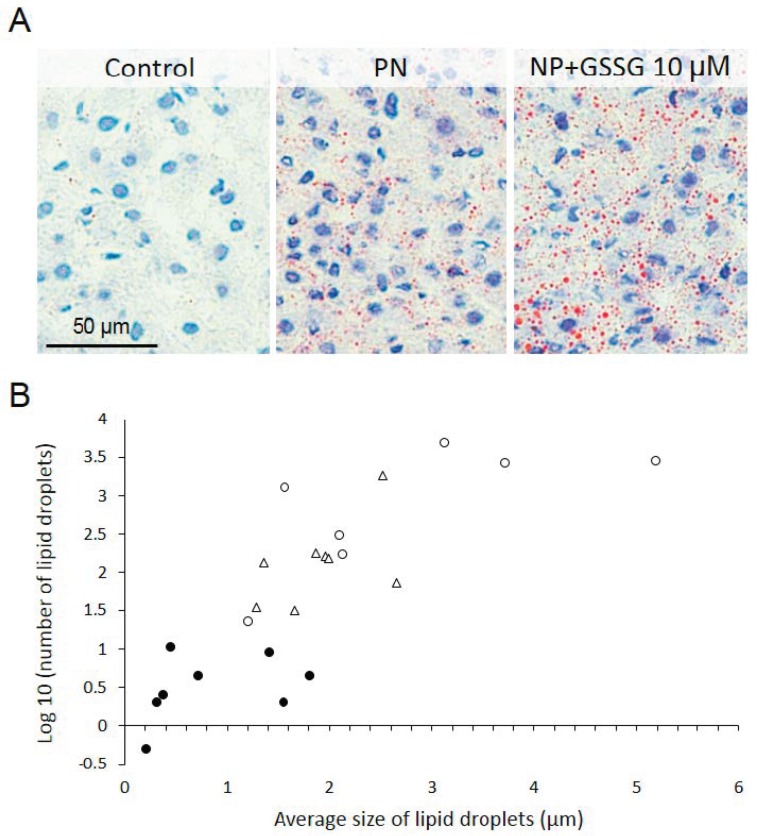
Evaluation of hepatic lipid content. (**A**) Representative picture of each studied group. (**B**) Lipid droplets in function of their number (log_10_), size (µm), and groups (Dark circle: Reference group. Open triangle: Parenteral nutrition (PN) group. Open circle: PN + 10 µM GSSG group). PN ± GSSG induced an increased (*p* < 0.01) number of hepatic lipid droplets.

**Figure 3 nutrients-11-02063-f003:**
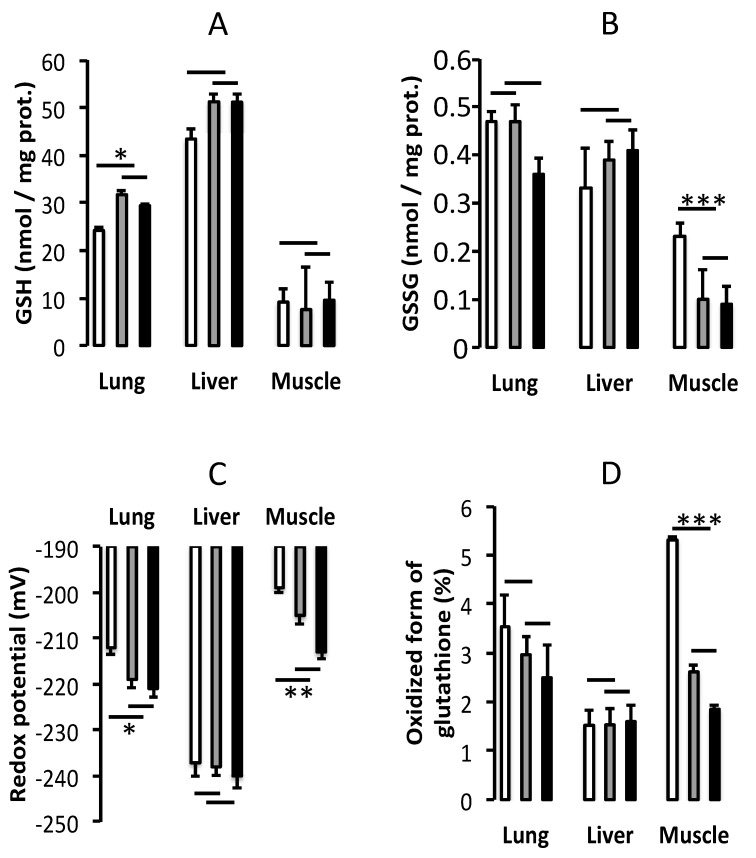
Glutathione. Reduced form of glutathione (GSH) (**A**), oxidized form of glutathione (GSSG) (**B**), redox potential (**C**), and proportion of glutathione in oxidized form (**D**) in lungs, liver, and gastrocnemius muscle. White column: PN (Parenteral nutrition). Gray column: PN + 10 µM GSSG. Black column: Reference group (animals fed orally). Data are expressed as mean ± S.E.M., *n* = 7–8 per group. The bars show the statistical comparisons. The absence of a symbol on the bar means that *p* > 0.05. *: *p* < 0.05; **: *p* < 0.01; ***: *p* < 0.001.

**Figure 4 nutrients-11-02063-f004:**
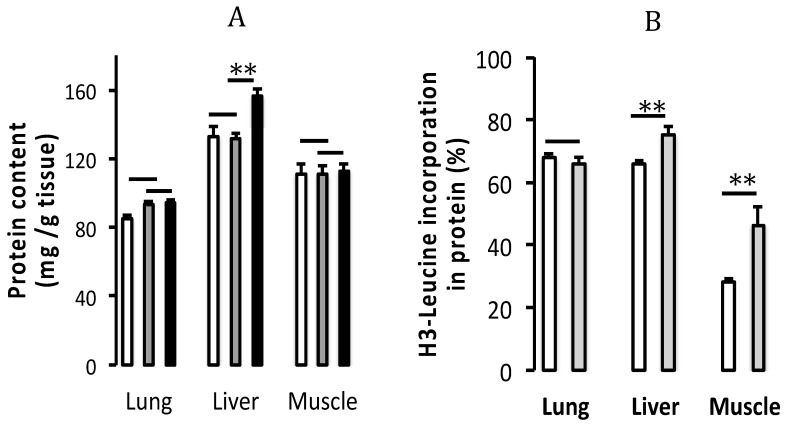
Protein content and index of protein synthesis. (**A**) Protein content (mg/g of tissue) and (**B**) index of protein synthesis (% of H^3^-leucine in the protein fraction) in lungs, liver and gastrocnemius muscle. White column: PN (Parenteral nutrition). Gray column: PN + 10 µM GSSG. Black column: Reference group (animals fed orally). Data are expressed as mean ± S.E.M., *n* = 7–8 per group. The bars show the statistical comparisons. The absence of a symbol on the bar means that *p* > 0.05. **: *p* < 0.01.

**Table 1 nutrients-11-02063-t001:** Mean daily caloric intake. Caloric intake are expressed as kcal/kg of body weight for each day on parenteral nutrition (PN) or PN enriched with GSSG. The daily caloric intake increased with time (F_(4,65)_ = 8.20, *p* < 0.001) reaching a plateau at the third day (F_(1,65)_ ≤ 2.3).

	Mean Daily Caloric Intake (kcal/kg)
Day 1	60 ± 3
Day 2	69 ± 4
Day 3	84 ± 4
Day 4	83 ± 5
Day 5	93 ± 6

**Table 2 nutrients-11-02063-t002:** Haemoglobin and plasma urea concentrations. Hb: haemoglobin measured on the first (d0) and last (d5) day of experimentation. Urea measured in plasma at the last day of experimentation. At d0, there was no difference in Hb between groups, while at d5 it was higher in PN group compared to PN+GSSG and reference groups. Urea concentrations were not significantly different between groups. Mean ± S.E.M., *n* = 4–8 per group; **: *p* < 0.01.

	PN	PN+10 µM GSSG	Reference Group
Hb at d0 (g/L)	198 ± 3	188 ± 4	192 ± 3
Hb at d5 (g/L)	213 ± 4 **	199 ± 5	191 ± 3
Urea at d5 (mg/L)	34.7 ± 1.3	34.3 ± 2.9	33.8 ± 1.0

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
