# Peer review of "Glutathione Supplementation of Parenteral Nutrition Prevents Oxidative Stress and Sustains Protein Synthesis in Guinea Pig Model"

_nutrients, 2019, doi:10.3390/nu11092063_

Round 1

Reviewer 1 Report

The aim of the study submitted by Morin G & al was to evaluate the effect of glutathione supplementation of PN on oxidative stress and protein synthesis in guinea pig model. It is an important issue especially for preterm infants. Indeed, we known from many years that cysteine is a semi-essential AA in preterm infants due to the immaturity of the transsulfuration pathway and that due to the reduced solubility of cystine in parenteral solution, the relative deficiency of parenteral AA solution cannot be compensate by an increase in methionine concentration. In addition, sulfur AA’s are important precursors of glutathione playing a major role in peroxide detoxification. In this study, a plasma determination of the sulfur AA concentration during the study could be of interest.

The present study requests some clarification and remarks.

Methods

1-There is a significant difference between the control group and the PN groups. Indeed, in the control group, the animals were housed in the animal facility for 5 days for acclimation and without any manipulation were fed with regular guinea pig food. By contrast, after 5 days of acclimation, the PN groups were anaesthetized to insert an external jugular vein catheter. During the first two days after surgery, the animals were fed ad libitum with regular guinea pig food and had free access to tap water. As a result, the manipulation induces in the PN groups a postnatal growth restriction accounting respectively to 6.5 and 5.1% at the beginning of the study period.

2- Metabolizable nutritional intakes differs between the control and the PN groups. The control group were fed ad libitum with regular guinea pig food and had free access to tap water without any evaluation of the protein and energy intakes. By contrast, the PN groups received a PN providing 60kcal/kg*d and 1.82 g AA/kg*d on day1 increasing progressively to 93 kcal/kg*d and 2.82 g AA/kg*d on day 5.

3-The authors suggest that the parenteral intakes are close to the recommendations in paediatrics parenteral nutrition (ref 27). In reference 27 it was suggested that ” In very low birth weight infants, to approximate intra-uterine lean body mass accretion and growth, energy intakes of 90-120 kcal/kg/day should be provided” but also that “After the initial postnatal nadir of weight loss, aiming for a weight gain of 17-20 g/kg per day in very low birth weight infants is recommended to prevent dropping across weight centiles i.e. growth failure”. Thus, this reference suggests that in preterm infants, a weight gain of 17 g/kg/d could be observed for the minimal energy intake of 90 kcal/kg*d. Unfortunately, no clinical study supports this affirmation. By contrast, several older studies showed that the energy cost of growth (ie, for synthesis of, and storage in, new tissue) was 4.9 kcal/gm of weight gain (Reichman, B L Pediatrics, 1982). According to these studies, 17g/kg/d requests 85 metabolizable kcal/kg*d for growth but an additional 50 kcal/kg*d for basal metabolism and thermoregulation.

4- Relative weight gain is higher in guinea pig than in preterm infants. According to the reference group (fig 1) weight gain during the study is 9.7 g/d corresponding to 25.8 g/kg*d suggesting that the optimal protein and energy intake could be significantly higher than that of preterm infants. By contrast to the control group, a negative weight gain was observed in the PN groups corresponding to -11.3 and -12.1 g/kg*d in the PN and the PN+GSSG groups respectively. Such a persistent weight loss suggests that booth groups remain in catabolic state during the study period that could be related to the post-surgical stress and/or to a cumulative protein and energy deficits.

5-Line 121 As expected, the PN groups were contaminated with 272 ± 14 μM peroxides [3,26]. This sentence need to be clarified Is it the result of the peroxide determination in the PN solution as suggested in line 129? They need to be presented in the result section. What mean the reference [3,26]? Results similar to the data presented in the 2 references 3 & 26.

6-PN inducing hepatic steatosis; Why did you chose to use Intralipid 20 for this experimentation and not a lipid emulsion with lesser toxicity such as the Clinoleic or the SMOFLipid? Combining the number of lipid droplets with their size, your study suggests that the combination of PN + GSSG increase the hepatic steatosis in the guinea pig. Did you test statistically the combination? Early development of liver steatose in animal receiving suboptimal nutritional intakes is surprizing. Did the authors had some explanations? Could you also speculate on the potential deleterious effect of the addition of 10µM of GSSG on liver fat deposition in the guinea pigs.

7-Validation of protein synthesis index in animals in negative nitrogen balance and catabolic state. As suggested before the relatively severe negative weight gain during the study period reported in the PN groups (-11,3 & -12.1 g/kg*J) by contrast to the large positive weight gain reported in the control group (25.8 g/kg*d) suggests that protein synthesis was negatively affected during the study period. The protein content per g of tissues were similar between the 2 PN group but lower than the control group in the liver. Unfortunately, the H3-Leucine incorporation test was not performed in the control group where a large positive protein synthesis could be suspected. Therefore we think that it is difficult to conclude from the data that the addition of 10 µM of GSSG could induce a significant improvement in protein synthesis in a group of animals in negative weight gain.

8 Sulphur metabolisms in guinea pig versus preterm infants. In preterm infants, there is a cystathionase relative deficiency that combined with a reduced solubility in parenteral solution potentialize the essential character of the cystine AA. Is the situation similar in the one-month old guinea pig? In preterm infants on total parenteral nutrition, Plasma methionine concentration is directly related to the intake By contrast plasma cysteine concentration is poorly related to intakes but directly related to the methionine intake. In addition, this relationship is also related to gestational age improving progressively from 28 to 36 postconceptional age suggesting a progressive maturation of the cystathionase activity. The ratio meth/cys decrease progressively. The cystathionase activity and its potential evolution according to postnatal age needs to be discussed in the manuscript

9-In conclusion, it is a very nice and promising study that could solve one of the major nutritional deficits in parenteral nutrition. But before the authors need to convince the reviewers that the surgical stress combined with a large nutritional deficit had no impact on the present results

Author Response

Answers to the Comments from reviewer #1:

The present study requests some clarification and remarks.

1) 1-There is a significant difference between the control group and the PN groups. Indeed, in the control group, the animals were housed in the animal facility for 5 days for acclimation and without any manipulation were fed with regular guinea pig food. By contrast, after 5 days of acclimation, the PN groups were anaesthetized to insert an external jugular vein catheter. During the first two days after surgery, the animals were fed ad libitum with regular guinea pig food and had free access to tap water. As a result, the manipulation induces in the PN groups a postnatal growth restriction accounting respectively to 6.5 and 5.1% at the beginning of the study period.

2- Metabolizable nutritional intakes differs between the control and the PN groups. The control group were fed ad libitum with regular guinea pig food and had free access to tap water without any evaluation of the protein and energy intakes. By contrast, the PN groups received a PN providing 60kcal/kg*d and 1.82 g AA/kg*d on day1 increasing progressively to 93 kcal/kg*d and 2.82 g AA/kg*d on day 5.

Answer to comments 1 and 2:

Indeed, there is a significant difference between the PN groups and the reference group. This difference was highly expected. The representation of our study groups in clinical situation includes: the PN with no GSSG supplementation that represent the “standard of care or control group”, the PN with GSSG supplementation that represents the “intervention group” and the ad libitum animal group that represents healthy individuals “reference group”. This statement has been added after the first paragraph in Methods section, page 6. The second modification in the revised manuscript was to bring the text in line with the description of the studied groups in the Methods section. Thus, the no intervention, ad libitum group was termed “reference group” throughout the manuscript.

As the reviewer points out, a major difference is in the metabolizable nutritional intakes. After a period of acclimatization, the caloric intake of the animals receiving PN increased gradually to reach a plateau after three days. This progression constituted also another acclimatization period. In addition, surgery to fix the catheter was by itself a metabolic stress, as suggested by a lower body weights (5-7%) compared to the reference group after 2 days of ad libitum food with regular guinea pig food. Thus, the experimental protocol consisted in comparing the PN+GSSG group to the PN group (which could be considered as the control group). Animals without manipulation were used as a reference to appreciate the extent of the impact of GSSG supplementation compared to healthy animals.

This aspect of the experimental design has been included as the second paragraph of the Discussion section, page 18. For text consistency, another paragraph has been moved.

3-The authors suggest that the parenteral intakes are close to the recommendations in paediatrics parenteral nutrition (ref 27). In reference 27 it was suggested that ” In very low birth weight infants, to approximate intra-uterine lean body mass accretion and growth, energy intakes of 90-120 kcal/kg/day should be provided” but also that “After the initial postnatal nadir of weight loss, aiming for a weight gain of 17-20 g/kg per day in very low birth weight infants is recommended to prevent dropping across weight centiles i.e. growth failure”. Thus, this reference suggests that in preterm infants, a weight gain of 17 g/kg/d could be observed for the minimal energy intake of 90 kcal/kg*d. Unfortunately, no clinical study supports this affirmation. By contrast, several older studies showed that the energy cost of growth (ie, for synthesis of, and storage in, new tissue) was 4.9 kcal/gm of weight gain (Reichman, B L Pediatrics, 1982). According to these studies, 17g/kg/d requests 85 metabolizable kcal/kg*d for growth but an additional 50 kcal/kg*d for basal metabolism and thermoregulation.

Answer

Reference 27 is a recommendation of the recent ESPGHAN/ESPEN/ESPR/CSPEN guidelines (2018). Thanks for the additional information from Reichman, B L Pediatrics, 1982. The following sentences have been added to the end of the third paragraph, page 19.

“Energy is an important factor for this synthesis. PN groups received energy intakes similar to those recommended in the recent (2018) ESPGHAN/ESPEN/ESPR/CSPEN guidelines for preterm newborns [27]. However, this recommendation does not take into account the energy required for thermoregulation, basal metabolism and activity (~50 kcal / kg / d) as suggested by Reichman BL et al [42]. The energy cost of these parameters is unknown in our animal model. Suboptimal energy intake can compromise protein synthesis. On the other hand, it may not be appropriate to compare the energy requirements of a month-old guinea pig to a premature newborn. The second important factor for this synthesis is the availability of amino acids, including cysteine. Here, GSSG has been used as pro-cysteine.”

4- Relative weight gain is higher in guinea pig than in preterm infants. According to the reference group (fig 1) weight gain during the study is 9.7 g/d corresponding to 25.8 g/kg*d suggesting that the optimal protein and energy intake could be significantly higher than that of preterm infants. By contrast to the control group, a negative weight gain was observed in the PN groups corresponding to -11.3 and -12.1 g/kg*d in the PN and the PN+GSSG groups respectively. Such a persistent weight loss suggests that booth groups remain in catabolic state during the study period that could be related to the post-surgical stress and/or to a cumulative protein and energy deficits.

Answer

Weight loss close to 6% (5.5 and 5.7% respectively for PN and PN+GSSG groups) during the PN period may suggest catabolism. However, the similarity of plasma concentrations of urea (the main nitrogen product of protein degradation) between groups does not support the assumption of the presence of catabolism. In addition, equal levels of protein per gram of muscle tissue in all groups, including reference group, suggest that PN animals did not suffer from inadequate caloric intake leading to higher protein catabolism. Despite apparently adequate caloric intake, reflected by haemoglobin levels and normal urea plasma concentrations, this lack of weight gain could be explained by qualitative and quantitative differences in metabolizable nutrients intake between PN groups and the reference group, or by the impact of the nutrient delivery route during growth. Another possibility is that this lack of growth in PN animals could be explained, at least in part, by a partial inability to de novo synthesize proteins. Suboptimal intake of cysteine is suspected. Here, GSSG has been used as pro-cysteine.

These arguments appear in the third paragraph of the Discussion section and have not been modified in the corrected version.

5-Line 121 As expected, the PN groups were contaminated with 272 ± 14 μM peroxides [3,26]. This sentence need to be clarified Is it the result of the peroxide determination in the PN solution as suggested in line 129? They need to be presented in the result section. What mean the reference [3,26]? Results similar to the data presented in the 2 references 3 & 26.

Answer

Thank you for this comment allowing us to provide clarification. This concentration of peroxides was measured in the PN solutions and was similar to levels previously measured in such solutions (references 3 and 26).

This result is now included in the “Oxidative stress” subsection of the Results section of the corrected version of the manuscript (page 13): “The PN solutions in the present study were contaminated with 272 ± 14 µM peroxide. This concentration was similar to levels previously measured in such solutions (references 3 and 26). This amount of peroxides has the potential to induce oxidative stress.“

6-PN inducing hepatic steatosis; Why did you chose to use Intralipid 20 for this experimentation and not a lipid emulsion with lesser toxicity such as the Clinoleic or the SMOFLipid? Combining the number of lipid droplets with their size, your study suggests that the combination of PN + GSSG increase the hepatic steatosis in the guinea pig. Did you test statistically the combination? Early development of liver steatose in animal receiving suboptimal nutritional intakes is surprizing. Did the authors had some explanations? Could you also speculate on the potential deleterious effect of the addition of 10µM of GSSG on liver fat deposition in the guinea pigs.

Answer

Intralipid was used because it remains widely used in all North American NICUs including our neonatal unit (Saint-Justine hospital). In a previous study whit same animal model of PN comparing the impact of PN compounding with SMOFLipid to a PN containing Intralipid, oxidative stress was greater in the lungs (Lavoie JC, Mohamed I, Nuyt AM, Elremaly W, Rouleau T. Impact of SMOFLipid on pulmonary alveolar development in newborn guinea pigs, JPEN J Parenter Enteral Nutr 42 :1314-1321, 2018) and in the liver (manuscript in preparation) of animals receiving SMOFLipid. Of course, it will be interesting to evaluate the impact of GSSG supplementation in animals receiving PN containing other lipid emulsions such as SMOFLipid and Clinoleic.

Indeed, the number of lipid droplets was higher in the PN+GSSG than in the PN group. The text has been modified in the Result section (page 11) to emphasize this statistical difference: “It (number of lipid droplets) was higher in the PN+GSSG group than in the PN group (F(1,18) = 8.13, p<0.05)”

Undeniably, it was surprising to observe more pronounced steatosis in PN+GSSG group. The direct impact of glutathione (GSH or GSSG) or redox potential is excluded because they were not influenced by PN or PN+GSSG (figure 3). A future dose-response study can confirm whether the difference observed is due to a statistical error of Type 1 or whether supplementation with GSSG induces or aggravates NP-induced steatosis. This statement has been added in the penultimate paragraph of the Discussion section in addition to including the concept of lipid accumulation in the liver as a parameter to be followed in a future dose-response study.

7-Validation of protein synthesis index in animals in negative nitrogen balance and catabolic state. As suggested before the relatively severe negative weight gain during the study period reported in the PN groups (-11,3 & -12.1 g/kg*J) by contrast to the large positive weight gain reported in the control group (25.8 g/kg*d) suggests that protein synthesis was negatively affected during the study period. The protein content per g of tissues were similar between the 2 PN group but lower than the control group in the liver. Unfortunately, the H3-Leucine incorporation test was not performed in the control group where a large positive protein synthesis could be suspected. Therefore we think that it is difficult to conclude from the data that the addition of 10 µM of GSSG could induce a significant improvement in protein synthesis in a group of animals in negative weight gain.

Answer

The animals received a solution of PN (±GSSG) enriched with H3-leucine for two days. It was impossible to replicate this infusion in the reference group. Here, we demonstrate the effect of GSSG supplementation compared to the PN control group.

This is why a future dose-response study is needed as indicated in the Discussion section (pages 17 and 22). The current outcome is important because it opens up the possibility of using GSSG as pro-cysteine in PN to improve tissue glutathione levels, reduce the oxidative stress associated with PN and possibly recover protein synthesis. Gain or loss of weight is complex. The validation that GSSG is a useful pro-cysteine for PN, is in itself an important concept.

8 Sulphur metabolisms in guinea pig versus preterm infants. In preterm infants, there is a cystathionase relative deficiency that combined with a reduced solubility in parenteral solution potentialize the essential character of the cystine AA. Is the situation similar in the one-month old guinea pig? In preterm infants on total parenteral nutrition, Plasma methionine concentration is directly related to the intake By contrast plasma cysteine concentration is poorly related to intakes but directly related to the methionine intake. In addition, this relationship is also related to gestational age improving progressively from 28 to 36 postconceptional age suggesting a progressive maturation of the cystathionase activity. The ratio meth/cys decrease progressively. The cystathionase activity and its potential evolution according to postnatal age needs to be discussed in the manuscript

Answer

Thank you for this opportunity to improve the manuscript. The following text has been added to the second paragraph of the Introduction section (pages 4-5). “The immaturity of cystathionase (the last enzyme of this cascade) in premature infants [21] could explain the low level of glutathione in this population. In addition, the peroxides generated in the PN solution inhibit methionine adenosyl transferase [11], the first enzyme of this same cascade. Thus, PN infusion results in decreased levels of GSH in the lungs and blood of animals, exacerbating oxidative stress [8,15]. The cystathionase activity increases with age [22] whereas the impact of peroxides seems independent of age. Indeed, in children (aged 1 to 10 years) receiving home-PN, the level of glutathione in blood is equal to half that of control children [23].”

9-In conclusion, it is a very nice and promising study that could solve one of the major nutritional deficits in parenteral nutrition. But before the authors need to convince the reviewers that the surgical stress combined with a large nutritional deficit had no impact on the present results

Answer

The purpose of the study was to evaluate the impact of GSSG supplementation of PN. As indicated in the answer to the first two comments, the comparison between PN and PN+GSSG isolates the GSSG supplementation from other PN characteristics. Of course, PN treatment includes surgery and a particular nutrition mode. The effect of GSSG supplementation is necessarily associated with these characteristics. For this reason, at the beginning of the Discussion section, the revised manuscript specifies which group is the control group (PN group).

Thank you for your comments. They helped us to improve our manuscript.

Reviewer 2 Report

Thanks for the opportunity to review. Here are my comments.

Page 1 (abstract). Research does not support the statement in lines 29-30. The results potentially support?

Page 1 line 18, change under to receiving

Page 2 line 38, change to nutrition support

Page 2 line 39 change impossible to not indicated

Page 2 line 45 change under to receiving

Page 5 line 167 change made possible to were utilized

Page 8 line 213 change induced to induce

Author Response

Answers to the Comments from reviewer #2:

Comments and Suggestions for Authors

Thanks for the opportunity to review. Here are my comments.

Page 1 (abstract). Research does not support the statement in lines 29-30. The results potentially support?

Answer: Thank you. “Potentially” has been added.

Page 1 line 18, change under to receiving

Answer: Thank you. The correction has been made.

Page 2 line 38, change to nutrition support

Answer: Thank you. The correction has been made.

Page 2 line 39 change impossible to not indicated

Answer: Thank you. The correction has been made.

Page 2 line 45 change under to receiving

Answer: Thank you. The correction has been made.

Page 5 line 167 change made possible to were utilized

Answer: Thank you. The correction has been made.

Page 8 line 213 change induced to induce

Answer: Thank you. The correction has been made.

Thank you for your comments. They helped us to improve our presentation of the manuscript.

Round 2

Reviewer 1 Report

We want to thank the authors for the improvement of their manuscript.

There is a few points that need to be clarified

Abstract and Introduction

Individuals under long-term PN are lighter and shorter than the average population.” It is true that limited growth has been suggested in long term parenteral nutrition during infancy. By contrast, in preterm infants and in VLBW infants on PN for a limited period, adequate growth have been reported following recent “more aggressive” parenteral nutrition. Therefore, it appears that although the parenteral AA solution are still deficient in sulphur (cysteine) and aromatic (tyrosine) AA, adequate nitrogen retention and growth could be observed. Nevertheless, these AA solutions induce a relative glutathione deficiency as well as an oxidative stress that could be deleterious.

About hepatic steatosis :

The revised manuscript suggests that the number of lipid droplets was similar in the reference group and in the PN group? But also that the Lipid droplet average size was also similar in the reference and in the PN groups.  At the end the conclusion could be  that the number and the size of the PN group were not significantly different from the reference group and that the presence of lipid in PN groups is limited to the PN+GSSG.

 Please Clarify

Although the total protein contents of whole blood in the three groups were not different (149±3 mg/mL; F(1,20)<1.8), they were higher (F(1,20)=15.4, p<0.001) in the plasma of animals receiving PN+ GSSG and in the reference group (60±1 mg/mL; F((1,20)=0.2) than in the plasma of the PN group (52±2 mg/mL).

Un fortunately, data on plasma protein at d0 were not determined and a potential positive effect of the GSSG supplementation can’t be speculate.

The PN was compounded with: 2% (w/v) amino acids preparation

Corresponding to 2g AA/100ml or 37,8 mg of L-cystein/100ml or 156µMol/100ml

In the intervention group the PN was supplemented with 10µmol of GSSG/100 ml.  Please confirm?

If I remember correctly the GSSG could provide 2 cystein molecules and therefore 20 µmol of cysteine increasing the relative cysteine content of the PN of 12.8%. According to that, addition of 10µmol of GSSG  account to about 20µmol of cysteine corresponding to about 1/7.5 of the cysteine concentration initially dissolved in the PN.

Comparisons of haemoglobin

According to table 2 Hgb levels were similar at inclusion and remain at the same level after the study period in the reference group but increase from 188 to 199 (p=?) in the experimental group and from 198 to 213 (p=?) in the PN group. Thus, the increases were similar in the 2 PN groups (+11 versus +15) and are probably related to the relative weight loss observed in the 2 groups. In the discussion it was stressed that “the haemoglobin levels and the normal urea plasma concentrations could reflect an apparently adequate caloric intake “. This sentence is very speculative. The haemoglobin turnover is not so fast to induce a significant change in 5 days that could be related to a protein or energy deficit. Similarly, the increase in Hgb concentrations observed during the 5 days don’t reflect protein synthesis but is probably more related to a relative haemoconcentration. Unfortunately, the protein and energy intakes of the healthy guinea pigs have not been recorded in the reference group.

Comparisons of plasma urea concentrations

With only a data after the 5 day of parenteral, it is very difficult to speculate on the adequacy of the protein energy intakes, the possible catabolic status or the protein synthesis during the study period.

Efficacy of the trans-sulphur pathway in guinea pig

Unfortunately, the maturity of the trans-sulphur pathway in the guinea pig was not discussed in the manuscript.  In the Primene AA solution, the methionine content was limited 24 mg of methionine and 19 mg of cysteine/g of AA in order to limit hypermethioninemia due to the immaturity of the cystathionase activity in preterm infants.

In the guinea pig, growth  rate is faster than that in preterm infants 25.8 g/kg*d in the reference group and the sulphur AA intakes could be limiting for protein synthesis in the 2 PN groups. According to the cystathionase activity, and by contrast to the preterm infants, an increase in the methionine supply could be satisfactory to correct the relative glutathione deficit observed in the PN groups.

“Here, the similarity of urea plasma concentrations between groups does not support the presence of catabolism”

It is not completely correct. At similar protein intake a increase in catabolism will induce an increase in urea. But in the present study the protein intake is higher in the reference group. Therefore you cannot conclude that the PN groups are not in catabolic state when they are losing body weight.   By contrast the similar urea concentration in the 2 PN groups don't support a significant effect of GSSG on protein synthesis. Indeed, a significant increase in protein synthesis would lead to a plasma urea reduction what is not observed in the present study. In addition, urea was only determined at 5 days and the change from day 0 was not evaluate.

We want to thank the authors for the improvement of their manuscript.

There is a few points that need to be clarified

Abstract and Introduction

Individuals under long-term PN are lighter and shorter than the average population.” It is true that limited growth has been suggested in long term parenteral nutrition during infancy. By contrast, in preterm infants and in VLBW infants on PN for a limited period, adequate growth have been reported following recent “more aggressive” parenteral nutrition. Therefore, it appears that although the parenteral AA solution are still deficient in sulphur (cysteine) and aromatic (tyrosine) AA, adequate nitrogen retention and growth could be observed. Nevertheless, these AA solutions induce a relative glutathione deficiency as well as an oxidative stress that could be deleterious.

About hepatic steatosis :

The revised manuscript suggests that the number of lipid droplets was similar in the reference group and in the PN group? But also that the Lipid droplet average size was also similar in the reference and in the PN groups.  At the end the conclusion could be  that the number and the size of the PN group were not significantly different from the reference group and that the presence of lipid in PN groups is limited to the PN+GSSG.

 Please Clarify

Although the total protein contents of whole blood in the three groups were not different (149±3 mg/mL; F(1,20)<1.8), they were higher (F(1,20)=15.4, p<0.001) in the plasma of animals receiving PN+ GSSG and in the reference group (60±1 mg/mL; F((1,20)=0.2) than in the plasma of the PN group (52±2 mg/mL).

Un fortunately, data on plasma protein at d0 were not determined and a potential positive effect of the GSSG supplementation can’t be speculate.

The PN was compounded with: 2% (w/v) amino acids preparation

Corresponding to 2g AA/100ml or 37,8 mg of L-cystein/100ml or 156µMol/100ml

In the intervention group the PN was supplemented with 10µmol of GSSG/100 ml.  Please confirm?

If I remember correctly the GSSG could provide 2 cystein molecules and therefore 20 µmol of cysteine increasing the relative cysteine content of the PN of 12.8%. According to that, addition of 10µmol of GSSG  account to about 20µmol of cysteine corresponding to about 1/7.5 of the cysteine concentration initially dissolved in the PN.

Comparisons of haemoglobin

According to table 2 Hgb levels were similar at inclusion and remain at the same level after the study period in the reference group but increase from 188 to 199 (p=?) in the experimental group and from 198 to 213 (p=?) in the PN group. Thus, the increases were similar in the 2 PN groups (+11 versus +15) and are probably related to the relative weight loss observed in the 2 groups. In the discussion it was stressed that “the haemoglobin levels and the normal urea plasma concentrations could reflect an apparently adequate caloric intake “. This sentence is very speculative. The haemoglobin turnover is not so fast to induce a significant change in 5 days that could be related to a protein or energy deficit. Similarly, the increase in Hgb concentrations observed during the 5 days don’t reflect protein synthesis but is probably more related to a relative haemoconcentration. Unfortunately, the protein and energy intakes of the healthy guinea pigs have not been recorded in the reference group.

Comparisons of plasma urea concentrations

With only a data after the 5 day of parenteral, it is very difficult to speculate on the adequacy of the protein energy intakes, the possible catabolic status or the protein synthesis during the study period.

Efficacy of the trans-sulphur pathway in guinea pig

Unfortunately, the maturity of the trans-sulphur pathway in the guinea pig was not discussed in the manuscript.  In the Primene AA solution, the methionine content was limited 24 mg of methionine and 19 mg of cysteine/g of AA in order to limit hypermethioninemia due to the immaturity of the cystathionase activity in preterm infants.

In the guinea pig, growth  rate is faster than that in preterm infants 25.8 g/kg*d in the reference group and the sulphur AA intakes could be limiting for protein synthesis in the 2 PN groups. According to the cystathionase activity, and by contrast to the preterm infants, an increase in the methionine supply could be satisfactory to correct the relative glutathione deficit observed in the PN groups.

“Here, the similarity of urea plasma concentrations between groups does not support the presence of catabolism”

It is not completely correct. At similar protein intake a increase in catabolism will induce an increase in urea. But in the present study the protein intake is higher in the reference group. Therefore you cannot conclude that the PN groups are not in catabolic state when they are losing body weight.   By contrast the similar urea concentration in the 2 PN groups don't support a significant effect of GSSG on protein synthesis. Indeed, a significant increase in protein synthesis would lead to a plasma urea reduction what is not observed in the present study. In addition, urea was only determined at 5 days and the change from day 0 was not evaluate.

Author Response

Answers to the Comments from reviewer #1:

We want to thank the authors for the improvement of their manuscript.

There is a few points that need to be clarified

Abstract and Introduction

“Individuals under long-term PN are lighter and shorter than the average population.” It is true that limited growth has been suggested in long term parenteral nutrition during infancy. By contrast, in preterm infants and in VLBW infants on PN for a limited period, adequate growth have been reported following recent “more aggressive” parenteral nutrition. Therefore, it appears that although the parenteral AA solution are still deficient in sulphur (cysteine) and aromatic (tyrosine) AA, adequate nitrogen retention and growth could be observed. Nevertheless, these AA solutions induce a relative glutathione deficiency as well as an oxidative stress that could be deleterious.

Answer: Of course, there is a difference between the impacts of the PN in the short-term and those in the long-term.

The second sentence of the abstract (“Thus, PN causes a cysteine deficiency, characterized by low levels of glutathione, the main molecule used in peroxide detoxification. Individuals receiving long-term PN are lighter and shorter than the average population”) has been replaced by: “Thus, PN causes a cysteine deficiency, characterized by low levels of glutathione, the main molecule used in peroxide detoxification, and limited growth in individuals receiving long-term PN compared to average population.”

In Introduction section, page 5, second paragraph, line 5, the following text has been added: “ … unlike preterm newborns who receive short-term PN for a limited time, …” just before the text: “studies reported that infants receiving long-term parenteral nutrition had a lighter weight [27] and a smaller height [27,28] than the population of the same age.”

About hepatic steatosis :

The revised manuscript suggests that the number of lipid droplets was similar in the reference group and in the PN group? But also that the Lipid droplet average size was also similar in the reference and in the PN groups.  At the end the conclusion could be  that the number and the size of the PN group were not significantly different from the reference group and that the presence of lipid in PN groups is limited to the PN+GSSG.

Answer: In the penultimate paragraph of the Discussion section (page 22), the sentence “It was surprising to observe more pronounced steatosis in the PN+GSSG group” has been replaced by the following: “Steatosis was expected in the PN groups, but it was surprising to observe more lipid droplets in the PN group containing GSSG than in the PN group without GSSG. The size of the droplets was not influenced by the presence of GSSG in the NP.”

 Please Clarify

Although the total protein contents of whole blood in the three groups were not different (149±3 mg/mL; F(1,20)<1.8), they were higher (F(1,20)=15.4, p<0.001) in the plasma of animals receiving PN+ GSSG and in the reference group (60±1 mg/mL; F((1,20)=0.2) than in the plasma of the PN group (52±2 mg/mL).

Un fortunately, data on plasma protein at d0 were not determined and a potential positive effect of the GSSG supplementation can’t be speculate.

Answer: We agree that data at d0 would further confirm the demonstration (paired measurements). The significant difference observed at d5 (unpaired measurements), like any statistically significant difference, can be due to a type 1 error. Therefore, as indicated in the fourth line of the discussion, these data, like all others, must be confirmed by a dose-response study.

On the other hand, this data should be considered as part of the overall results observed with GSSG. For example, compared to PN, the higher plasma protein content of the PN+GSSG group is well aligned with the higher protein synthesis index in the liver. We can speculate a higher production of albumin. The discussion does not elaborate on this observation, leaving the reader to estimate for himself the importance of this data. No change has been made to the new version of the manuscript.

The PN was compounded with: 2% (w/v) amino acids preparation

Corresponding to 2g AA/100ml or 37,8 mg of L-cystein/100ml or 156µMol/100ml

In the intervention group the PN was supplemented with 10µmol of GSSG/100 ml.  Please confirm?

Answer: In the intervention group the PN was supplement with 10 µmol of GSSG / liter (10µM).

If I remember correctly the GSSG could provide 2 cystein molecules and therefore 20 µmol of cysteine increasing the relative cysteine content of the PN of 12.8%. According to that, addition of 10µmol of GSSG  account to about 20µmol of cysteine corresponding to about 1/7.5 of the cysteine concentration initially dissolved in the PN.

Answer: Indeed, GSSG could provide 2 cysteines. 10 µmol of GSSG / liter could provide 20 µmol cysteines / liter, or 2 µmol cysteines / 100 ml. 2% (w,v) Primene = 2g amino acids /100 ml = 0.31 mmol cysteine/100 ml (310 µmol cysteines / 100 ml). Therefore, GSSG added in PN could account for a maximum of 0.65% of the cysteine concentration initially dissolved in the PN.

Thank you for this attention. On the first line of the page 20, “about one fifteenth” has been changed for “less than one percent “

Comparisons of haemoglobin

According to table 2 Hgb levels were similar at inclusion and remain at the same level after the study period in the reference group but increase from 188 to 199 (p=?) in the experimental group and from 198 to 213 (p=?) in the PN group. Thus, the increases were similar in the 2 PN groups (+11 versus +15) and are probably related to the relative weight loss observed in the 2 groups. In the discussion it was stressed that “the haemoglobin levels and the normal urea plasma concentrations could reflect an apparently adequate caloric intake “. This sentence is very speculative. The haemoglobin turnover is not so fast to induce a significant change in 5 days that could be related to a protein or energy deficit. Similarly, the increase in Hgb concentrations observed during the 5 days don’t reflect protein synthesis but is probably more related to a relative haemoconcentration. Unfortunately, the protein and energy intakes of the healthy guinea pigs have not been recorded in the reference group.

Answer: The difference between d0 and d5 in PN+GSSG was not significant, p > 0.05. The difference between d0 and d5 in PN was significant, p < 0.01.

We agree that, depending on the life span of Hb, the modification can not be linked to its synthesis. As noted in the Methods section on page 9, “ haemoglobin … values were utilized to assess the animal overall health such as anemia, dehydration and starvation”.

We agree with your comment. In page 18, second paragraph, line 9, the following text “reflected by haemoglobin levels and normal urea plasma concentrations” has been deleted from the new version of the manuscript.

Comparisons of plasma urea concentrations

With only a data after the 5 day of parenteral, it is very difficult to speculate on the adequacy of the protein energy intakes, the possible catabolic status or the protein synthesis during the study period.

Answer: Thank you for your comment.

Initially, as described in the Methods section on page 9, “ haemoglobin and plasma urea values were utilized to assess the animal overall health such as anemia, dehydration and starvation. The data collected on d5 seemed sufficient. Since the value of urea is also affected by the state of catabolism (starvation), it has also served as an index of catabolism. Thus, in absence of its modification, the catabolism was not apparent. This is why the text on the line 6 of the second paragraph, page 18 of the Discussion section, specifies that the data “do not support the presence of catabolism” without excluding it.

Efficacy of the trans-sulphur pathway in guinea pig

Unfortunately, the maturity of the trans-sulphur pathway in the guinea pig was not discussed in the manuscript.  In the Primene AA solution, the methionine content was limited 24 mg of methionine and 19 mg of cysteine/g of AA in order to limit hypermethioninemia due to the immaturity of the cystathionase activity in preterm infants.

In the guinea pig, growth  rate is faster than that in preterm infants 25.8 g/kg*d in the reference group and the sulphur AA intakes could be limiting for protein synthesis in the 2 PN groups. According to the cystathionase activity, and by contrast to the preterm infants, an increase in the methionine supply could be satisfactory to correct the relative glutathione deficit observed in the PN groups.

Answer: The animal model used in our study was one month old. At this age, we expect the trans-sulphuration pathway to be mature. However, the inhibition of the first enzyme of this pathway by the peroxides contaminating PN is independent of age because it is caused by chemical oxidation of the cysteinyl residues of methionine adenosyltransferase (see Introduction, pages 4-5). In case of a partial inhibition, the increase of the concentration of the substrate methionine could lead to an increase synthesis of cysteine / glutathione. This hypothesis deserves to be tested. A protocol comparing PN compounded with Travasol (higher methionine, no cysteine) rather than Primene could answer the question. No modification has been made to the manuscript.

“Here, the similarity of urea plasma concentrations between groups does not support the presence of catabolism”

It is not completely correct. At similar protein intake a increase in catabolism will induce an increase in urea. But in the present study the protein intake is higher in the reference group. Therefore you cannot conclude that the PN groups are not in catabolic state when they are losing body weight.   By contrast the similar urea concentration in the 2 PN groups don't support a significant effect of GSSG on protein synthesis. Indeed, a significant increase in protein synthesis would lead to a plasma urea reduction what is not observed in the present study. In addition, urea was only determined at 5 days and the change from day 0 was not evaluate.

Answer: As specified in a previous answer, the urea data do not support an apparent catabolism, but do not exclude it. The amount of GSSG added in PN was based on previous study demonstrating that 10 uM GSSG in PN was sufficient to prevent oxidative stress and alveoli loss in lungs of neonatal guinea pigs receiving PN for 4 days (see Methods section, Experimental design, last sentence of the second paragraph, page 7). Protection against oxidative stress has also been a successful here in lungs and muscles.

The protein synthesis index used here also suggests the utility of adding GSSG in the PN. In the first paragraph of the Discussion section, it is clearly state that these results including improved protein synthesis, need to be confirmed by a dose-response study. As specified in the manuscript, the amount of cysteine provided by GSSG is small and can be improved. The best indicator for the optimal dose of GSSG will probably be plasma glutathione, for which the normal value is known. Here, this normal value has not been reached.

Thank you for your comments. They helped us to improve our presentation of the manuscript.
